# Learnable Behavior Control: Breaking Atari Human World Records via Sample-Efficient Behavior Selection

**Jiajun Fan**[1][*]**, Yuzheng Zhuang**[2] **, Yuecheng Liu**[2] **, Jianye Hao**[2] **, Bin Wang**[2]

**Jiangcheng Zhu**[3]**, Hao Wang**[4] **, Shutao Xia**[1] [†]

[1] Tsinghua Shenzhen International Graduate School, Tsinghua University
[2] Huawei Noah's Ark Lab, [3] Huawei Cloud, [4] Zhejiang University
[1] fanjj21@mails.tsinghua.edu.cn, xiast@sz.tsinghua.edu.cn, [4] haohaow@zju.edu.cn,
[2,3] {zhuangyuzheng, liuyuecheng1, haojianye, wangbin158, zhujiangcheng}@huawei.com

## Abstract

The exploration problem is one of the main challenges in deep reinforcement learning (RL). Recent promising works tried to handle the problem with population-based methods, which collect samples with diverse behaviors derived from a population of different exploratory policies. Adaptive policy selection has been adopted for behavior control. However, the behavior selection space is largely limited by the predefined policy population, which further limits behavior diversity. In this paper, we propose a general framework called **L**earnable **B**ehavioral **C**ontrol (LBC) to address the limitation, which a) enables a significantly enlarged behavior selection space via formulating a *hybrid behavior mapping* from all policies; b) constructs a unified *learnable process* for behavior selection. We introduce LBC into distributed off-policy actor-critic methods and achieve behavior control via optimizing the selection of the behavior mappings with bandit-based meta-controllers. Our agents have achieved 10077.52% mean human normalized score and surpassed 24 human world records within 1B training frames in the Arcade Learning Environment, which demonstrates our significant state-of-the-art (SOTA) performance without degrading the sample efficiency.

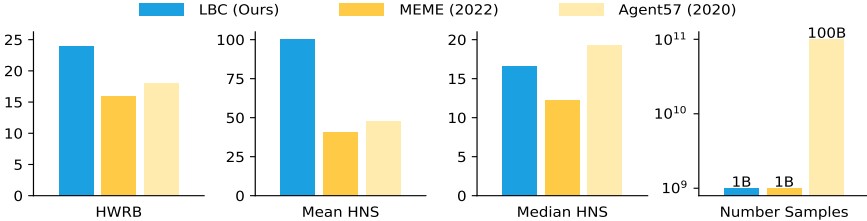

Figure 1: Performance on the 57 Atari games. Our method achieves the highest mean human normalized scores (Badia et al., 2020a), is the first to breakthrough 24 human world records (Toromanoff et al., 2019), and demands the least training data.

## 1 Introduction

Reinforcement learning (RL) has led to tremendous progress in a variety of domains ranging from video games (Mnih et al., 2015) to robotics (Schulman et al., 2015; 2017). However, efficient exploration remains one of the significant challenges. Recent prominent works tried to address the problem with population-based training (Jaderberg et al., 2017, PBT) wherein a population of policies with different degrees of exploration is jointly trained to keep both the long-term and short-term exploration capabilities throughout the learning process. A set of actors is created to acquire

---

[*]Work done as a research intern at Huawei Noah's Ark Lab.
[†]Corresponding authors.

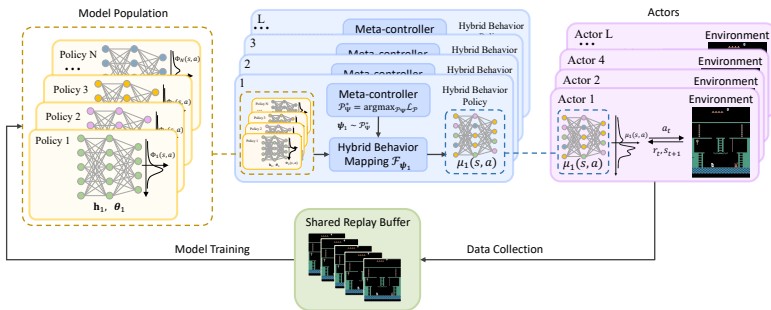

Figure 2: A General Architecture of Our Algorithm.

diverse behaviors derived from the policy population (Badia et al., 2020b;a). Despite the significant improvement in the performance, these methods suffer from the aggravated high sample complexity due to the joint training on the whole population while keeping the diversity property. To acquire diverse behaviors, NGU (Badia et al., 2020b) uniformly selects policies in the population regardless of their contribution to the learning progress (Badia et al., 2020b). As an improvement, Agent57 adopts an adaptive policy selection mechanism that each behavior used for sampling is periodically selected from the population according to a meta-controller (Badia et al., 2020a). Although Agent57 achieved significantly better results on the Arcade Learning Environment (ALE) benchmark, it costs tens of billions of environment interactions as much as NGU. To handle this drawback, GDI (Fan & Xiao, 2022) adaptively combines multiple advantage functions learned from a single policy to obtain an enlarged behavior space without increasing policy population size. However, the population-based scenarios with more than one learned policy has not been widely explored yet. Taking a further step from GDI, we try to enable a larger and non-degenerate behavior space by learning different combinations across a population of different learned policies.

In this paper, we attempt to further improve the sample efficiency of population-based reinforcement learning methods by taking a step towards a more challenging setting to control behaviors with significantly enlarged behavior space with a population of different learned policies. Differing from all of the existing works where each behavior is derived from a single selected learned policy, we formulate the process of getting behaviors from all learned policies as *hybrid behavior mapping*, and the behavior control problem is directly transformed into selecting appropriate mapping functions. By combining all policies, the behavior selection space increases exponentially along with the population size. As a special case that *population size degrades to one*, diverse behaviors can also be obtained by choosing different behavior mappings. This two-fold mechanism enables tremendous larger space for behavior selection. By properly parameterizing the mapping functions, our method enables a unified learnable process, and we call this general framework **L**earnable **B**ehavior **C**ontrol.

We use the Arcade Learning Environment (ALE) to evaluate the performance of the proposed methods, which is an important testing ground that requires a broad set of skills such as perception, exploration, and control (Badia et al., 2020a). Previous works use the normalized human score to summarize the performance on ALE and claim superhuman performance (Bellemare et al., 2013). However, the human baseline is far from representative of the best human player, which greatly underestimates the ability of humanity. In this paper, we introduce a more challenging baseline, *i.e.*, the human world records baseline (see Toromanoff et al. (2019); Hafner et al. (2021) for more information on Atari human world records). We summarize the number of games that agents can outperform the human world records (*i.e.*, HWRB, see Figs. 1) to claim a real superhuman performance in these games, inducing a more challenging and fair comparison with human intelligence. Experimental results show that the sample efficiency of our method also outperforms the concurrent work MEME Kapturowski et al. (2022), which is 200x faster than Agent57. In summary, our contributions are as follows:

1. **A data-efficient RL framework named LBC**. We propose a general framework called **L**earnable **B**ehavior **C**ontrol (LBC), which enables a significantly enlarged behavior selection space without increasing the policy population size via formulating a hybrid behavior mapping from all policies, and constructs a unified learnable process for behavior selection.

2. **A family of LBC-based RL algorithms**. We provide a family of LBC-based algorithms by combining LBC with existing distributed off-policy RL algorithms, which shows the generality and scalability of the proposed method.

3. **The state-of-the-art performance with superior sample efficiency**. From Figs. 1, our method has achieved 10077.52% mean human normalized score (HNS) and surpassed 24

human world records within 1B training frames in the Arcade Learning Environment (ALE), which demonstrates our state-of-the-art (SOTA) sample efficiency.

## 2 BACKGROUND

### 2.1 REINFORCEMENT LEARNING

RL can be formulated as a Markov Decision Process (Howard, 1960, MDP) defined by $(\mathcal{S}, \mathcal{A}, p, r, \gamma, \rho_0)$. Considering a discounted episodic MDP, the initial state $s_0$ is sampled from the initial distribution $\rho_0(s) : \mathcal{S} \to \mathbb{P}(\mathcal{S})$, where we use $\mathbb{P}$ to represent the probability distribution. At each time $t$, the agent chooses an action $a_t \in \mathcal{A}$ according to the policy $\pi(a_t|s_t) : \mathcal{S} \to \mathbb{P}(\mathcal{A})$ at state $s_t \in \mathcal{S}$. The environment receives $a_t$, produces the reward $r_t \sim r(s, a) : \mathcal{S} \times \mathcal{A} \to \mathbf{R}$ and transfers to the next state $s_{t+1}$ according to the transition distribution $p(s' \mid s, a) : \mathcal{S} \times \mathcal{A} \to \mathbb{P}(\mathcal{S})$. The process continues until the agent reaches a terminal state or a maximum time step. Define the discounted state visitation distribution as $d_{\rho_0}^{\pi}(s) = (1 - \gamma)\mathbb{E}_{s_0 \sim \rho_0}\left[\sum_{t=0}^{\infty} \gamma^t \mathbf{P}(s_t = s|s_0)\right]$. Define return $G_t = \sum_{k=0}^{\infty} \gamma^k r_{t+k}$ wherein $\gamma \in (0, 1)$ is the discount factor. The goal of reinforcement learning is to find the optimal policy $\pi^*$ that maximizes the expected sum of discounted rewards $G_t$:

$$\pi^* := \operatorname*{argmax}_{\pi} \mathbb{E}_{s_t \sim d_{\rho_0}^{\pi}} \mathbb{E}_{\pi}\left[G_t = \sum_{k=0}^{\infty} \gamma^k r_{t+k}|s_t\right], \tag{1}$$

### 2.2 BEHAVIOR CONTROL FOR REINFORCEMENT LEARNING

In value-based methods, a behavior policy can be derived from a state-action value function $Q_{\boldsymbol{\theta},\mathbf{h}}^{\pi}(s, a)$ via $\epsilon$-greedy. In policy-based methods, a behavior policy can be derived from the policy logits $\Phi_{\boldsymbol{\theta},\mathbf{h}}$ (Li et al., 2018) via Boltzmann operator. For convenience, we define that a behavior policy can be derived from the learned policy model $\Phi_{\boldsymbol{\theta},\mathbf{h}}$ via a behavior mapping, which normally maps a single policy model to a behavior, $e.g.$, $\epsilon$-greedy($\Phi_{\boldsymbol{\theta},\mathbf{h}}$). In PBT-based methods, there would be a set of policy models $\{\Phi_{\boldsymbol{\theta}_1,\mathbf{h}_1}, ..., \Phi_{\boldsymbol{\theta}_N,\mathbf{h}_N}\}$, each of which is parameterized by $\boldsymbol{\theta}_i$ and trained under its own hyper-parameters $\mathbf{h}_i$, wherein $\boldsymbol{\theta}_i \in \boldsymbol{\Theta} = \{\boldsymbol{\theta}_1, ..., \boldsymbol{\theta}_N\}$ and $\mathbf{h}_i \in \mathbf{H} = \{\mathbf{h}_1, ..., \mathbf{h}_N\}$.

The behavior control in population-based methods is normally achieved in two steps: i) select a policy model $\Phi_{\boldsymbol{\theta},\mathbf{h}}$ from the population. ii) applying a behavior mapping to the selected policy model. When the behavior mapping is rule-based for each actor ($e.g.$, $\epsilon$-greedy with rule-based $\epsilon$ ), the behavior control can be transformed into the policy model selection (See Proposition 1). Therefore, the optimization of the selection of the policy models becomes one of the critical problems in achieving effective behavior control. Following the literature on PBRL, NGU adopts a uniform selection, which is unoptimized and inefficient. Built upon NGU, Agent57 adopts a meta-controller to adaptively selected a policy model from the population to generate the behavior for each actor, which is implemented by a non-stationary multi-arm bandit algorithm. However, the policy model selection requires maintaining a large number of different policy models, which is particularly data-consuming since each policy model in the population holds heterogeneous training objectives.

To handle this problem, recent notable work GDI-H[3] (Fan & Xiao, 2022) enables to obtain an enlarged behavior space via adaptively controls the temperature of the softmax operation over the weighted advantage functions. However, since the advantage functions are derived from the same target policy under different reward scales, the distributions derived from them may tend to be similar (e.g., See App. N), thus would lead to degradation of the behavior space. Differing from all of the existing works where each behavior is derived from a single selected learned policy, in this paper, we try to handle this problem via three-fold: i) we bridge the relationship between the learned policies and each behavior via a hybrid behavior mapping, ii) we propose a general way to build a non-degenerate large behavior space for population-based methods in Sec. 4.1, iii) we propose a way to optimize the hybrid behavior mappings from a population of different learned models in Proposition. 2.

## 3 LEARNABLE BEHAVIOR CONTROL

In this section, we first formulate the behavior control problem and decouple it into two sub-problems: behavior space construction and behavior selection. Then, we discuss how to construct the behavior

space and select behaviors based on the formulation. By integrating both, we can obtain a general framework to achieve behavior control in RL, called learnable behavior control (LBC).

## 3.1 BEHAVIOR CONTROL FORMULATION

**Behavior Mapping**    Define *behavior mapping* $\mathcal{F}$ as a mapping from some policy model(s) to a behavior. In previous works, a behavior policy is typically obtained using a single policy model. In this paper, as a generalization, we define two kinds of $\mathcal{F}$ according to how many policy models they take as input to get a behavior. The first one, *individual behavior mapping*, is defined as a mapping from a single model to a behavior that is widely used in prior works, *e.g.*, $\epsilon$-greedy and Boltzmann Strategy for discrete action space and Gaussian Strategy for continuous action space; And the second one, *hybrid behavior mapping*, is defined to map all policy models to a single behavior, *i.e.*, $\mathcal{F}(\Phi_{\boldsymbol{\theta}_1, \mathbf{h}_1}, ..., \Phi_{\boldsymbol{\theta}_\mathrm{N}, \mathbf{h}_\mathrm{N}})$. The hybrid behavior mapping enables us to get a hybrid behavior by combining all policies together, which provides a greater degree of freedom to acquire a larger behavior space. For any behavior mapping $\mathcal{F}_{\boldsymbol{\psi}}$ parameterized by $\boldsymbol{\psi}$, there exists a family of behavior mappings $\mathcal{F}_{\boldsymbol{\Psi}} = \{\mathcal{F}_{\boldsymbol{\psi}} | \boldsymbol{\psi} \in \boldsymbol{\Psi}\}$ that hold the same parametrization form with $\mathcal{F}_{\boldsymbol{\psi}}$, where $\boldsymbol{\Psi} \subseteq \mathbf{R}^k$ is a parameter set that contains all possible parameter $\boldsymbol{\psi}$.

**Behavior Formulation**    As described above, in our work, a behavior can be acquired by applying a behavior mapping $\mathcal{F}_{\boldsymbol{\psi}}$ to some policy model(s). For the individual behavior mapping case, a behavior can be formulated as $\mu_{\boldsymbol{\theta}, \mathbf{h}, \boldsymbol{\psi}} = \mathcal{F}_{\boldsymbol{\psi}}(\Phi_{\boldsymbol{\theta}, \mathbf{h}})$, which is also the most used case in previous works. As for the hybrid behavior mapping case, a behavior is formulated as $\mu_{\boldsymbol{\Theta}, \mathbf{H}, \boldsymbol{\psi}} = \mathcal{F}_{\boldsymbol{\psi}}(\boldsymbol{\Phi}_{\boldsymbol{\Theta}, \mathbf{H}})$, wherein $\boldsymbol{\Phi}_{\boldsymbol{\Theta}, \mathbf{H}} = \{\Phi_{\boldsymbol{\theta}_1, \mathbf{h}_1}, ..., \Phi_{\boldsymbol{\theta}_\mathrm{N}, \mathbf{h}_\mathrm{N}}\}$ is a policy model set containing all policy models.

**Behavior Control Formulation**    Behavior control can be decoupled into two sub-problems: 1) which behaviors can be selected for each actor at each training time, namely the *behavior space construction*. 2) how to select proper behaviors, namely the *behavior selection*. Based on the behavior formulation, we can formulate these sub-problems:

**Definition 3.1** (Behavior Space Construction). *Considering the RL problem that behaviors $\mu$ are generated from some policy model(s). We can acquire a family of realizable behaviors by applying a family of behavior mappings $\mathcal{F}_{\boldsymbol{\Psi}}$ to these policy model(s). Define the set that contains all of these realizable behaviors as the behavior space, which can be formulated as:*

$$\mathbf{M}_{\boldsymbol{\Theta}, \mathbf{H}, \boldsymbol{\Psi}} = \begin{cases} \{\mu_{\boldsymbol{\theta}, \mathbf{h}, \boldsymbol{\psi}} = \mathcal{F}_{\boldsymbol{\psi}}(\Phi_{\mathbf{h}}) | \boldsymbol{\theta} \in \boldsymbol{\Theta}, \mathbf{h} \in \mathbf{H}, \boldsymbol{\psi} \in \boldsymbol{\Psi}\}, & \text{for individual behavior mapping} \\ \{\mu_{\boldsymbol{\Theta}, \mathbf{H}, \boldsymbol{\psi}} = \mathcal{F}_{\boldsymbol{\psi}}(\boldsymbol{\Phi}_{\boldsymbol{\Theta}, \mathbf{H}}) | \boldsymbol{\psi} \in \boldsymbol{\Psi}\}, & \text{for hybrid behavior mapping} \end{cases}$$

(2)

**Definition 3.2** (Behavior Selection). *Behavior selection can be formulated as finding a optimal selection distribution $\mathcal{P}^*_{\mathbf{M}_{\boldsymbol{\Theta}, \mathbf{H}, \boldsymbol{\Psi}}}$ to select the behaviors $\mu$ from behavior space $\mathbf{M}_{\boldsymbol{\Theta}, \mathbf{H}, \boldsymbol{\Psi}}$ and maximizing some optimization target $\mathcal{L}_{\mathcal{P}}$, wherein $\mathcal{L}_{\mathcal{P}}$ is the optimization target of behavior selection:*

$$\mathcal{P}^*_{\mathbf{M}_{\boldsymbol{\Theta}, \mathbf{H}, \boldsymbol{\Psi}}} := \underset{\mathcal{P}_{\mathbf{M}_{\boldsymbol{\Theta}, \mathbf{H}, \boldsymbol{\Psi}}}}{\operatorname{argmax}} \mathcal{L}_{\mathcal{P}}$$

(3)

## 3.2 BEHAVIOR SPACE CONSTRUCTION

In this section, we further simplify the equation 2, and discuss how to construct the behavior space.

**Assumption 1.** *Assume all policy models share the same network structure, and $\mathbf{h}_i$ can uniquely index a policy model $\Phi_{\boldsymbol{\theta}_i, \mathbf{h}_i}$. Then, $\Phi_{\boldsymbol{\theta}, \mathbf{h}}$ can be abbreviated as $\Phi_{\mathbf{h}}$.*

Unless otherwise specified, in this paper, we assume Assumption 1 holds. Under Assumption 1, the behavior space defined in equation 2 can be simplified as,

$$\mathbf{M}_{\mathbf{H}, \boldsymbol{\Psi}} = \begin{cases} \{\mu_{\mathbf{h}, \boldsymbol{\psi}} = \mathcal{F}_{\boldsymbol{\psi}}(\Phi_{\mathbf{h}}) | \mathbf{h} \in \mathbf{H}, \boldsymbol{\psi} \in \boldsymbol{\Psi}\}, & \text{for individual behavior mapping} \\ \{\mu_{\mathbf{H}, \boldsymbol{\psi}} = \mathcal{F}_{\boldsymbol{\psi}}(\boldsymbol{\Phi}_{\mathbf{H}}) | \boldsymbol{\psi} \in \boldsymbol{\Psi}\}, & \text{for hybrid behavior mapping} \end{cases}$$

(4)

According to equation 4, four core factors need to be considered when constructing a behavior space: the network structure $\Phi$, the form of behavior mapping $\mathcal{F}$, the hyper-parameter set $\mathbf{H}$ and the parameter set $\boldsymbol{\Psi}$. Many notable representation learning approaches have explored how to design the

network structure (Chen et al., 2021; Irie et al., 2021), but it is not the focus of our work. In this paper, we do not make any assumptions about the model structure, which means it can be applied to *any* model structure. Hence, there remains three factors, which will be discussed below.

For cases that behavior space is constructed with *individual behavior mappings*, there are two things to be considered if one want to select a specific behavior from the behavior space: the policy model $\Phi_{\mathbf{h}}$ and behavior mapping $\mathcal{F}_{\psi}$. Prior methods have tried to realize behavior control via selecting a policy model $\Phi_{\mathbf{h}_i}$ from the population $\{\Phi_{\mathbf{h}_1}, ..., \Phi_{\mathbf{h}_N}\}$ (See Proposition 1). The main drawback of this approach is that only one policy model is considered to generate behavior, leaving other policy models in the population unused. In this paper, we argue that we can tackle this problem via *hybrid behavior mapping*, wherein the hybrid behavior is generated based on all policy models.

In this paper, we only consider the case that all of the N policy models are used for behavior generating, *i.e.*, $\mu_{\mathbf{H},\psi} = \mathcal{F}_{\psi}(\Phi_{\mathbf{H}})$. Now there is only one thing to be considered , *i.e.*, the behavior mapping function $\mathcal{F}_{\psi}$, and the behavior control problem will be transformed into the optimization of the behavior mapping (See Proposition 2). We also do not make any assumptions about the form of the mapping. As an example, one could acquire a hybrid behavior from all policy models via network distillation, parameter fusion, mixture models, etc.

### 3.3 Behavior Selection

According to equation 4, each behavior can be indexed by $\mathbf{h}$ and $\psi$ for individual behavior mapping cases, and when the $\psi$ is not learned for each actor, the behavior selection can be cast to the selection of $\mathbf{h}$ (see Proposition 1). As for the hybrid behavior mapping cases, since each behavior can be indexed by $\psi$, the behavior selection can be cast into the selection of $\psi$ (see Proposition 2). Moreover, according to equation 3, there are two keys in behavior selection: **1)** Optimization Target $\mathcal{L}_{\mathcal{P}}$. **2)** The optimization algorithm to learn the selection distribution $\mathcal{P}_{\mathbf{M}_{\mathbf{H},\Psi}}$ and maximize $\mathcal{L}_{\mathcal{P}}$. In this section, we will discuss them sequentially.

**Optimization Target** Two core factors have to be considered for the optimization target: the diversity-based measurement $V_{\mu}^{\mathrm{TD}}$ (Eysenbach et al., 2019) and the value-based measurement $V_{\mu}^{\mathrm{TV}}$ (Parker-Holder et al., 2020). By integrating both, the optimization target can be formulated as:

$$\mathcal{L}_{\mathcal{P}} = \mathcal{R}_{\mu \sim \mathcal{P}_{\mathbf{M}_{\mathbf{H},\Psi}}} + c \cdot \mathcal{D}_{\mu \sim \mathcal{P}_{\mathbf{M}_{\mathbf{H},\Psi}}}$$
$$= \mathbb{E}_{\mu \sim \mathcal{P}_{\mathbf{M}_{\mathbf{H},\Psi}}}[V_{\mu}^{\mathrm{TV}} + c \cdot V_{\mu}^{\mathrm{TD}}], \tag{5}$$

wherein, $\mathcal{R}_{\mu \sim \mathcal{P}_{\mathbf{M}_{\mathbf{H},\Psi}}}$ and $\mathcal{D}_{\mu \sim \mathcal{P}_{\mathbf{M}_{\mathbf{H},\Psi}}}$ is the expectation of value and diversity of behavior $\mu$ over the selection distribution $\mathcal{P}_{\mathbf{M}_{\mathbf{H},\Psi}}$. When $\mathcal{F}_{\psi}$ is *unlearned* and *deterministic* for each actor, behavior selection for *each actor* can be simplified into the selection of the policy model:

**Proposition 1** (Policy Model Selection). *When $\mathcal{F}_{\psi}$ is a deterministic and individual behavior mapping for each actor at each training step (wall-clock), e.g., **Agent57**, the behavior for each actor can be uniquely indexed by $\mathbf{h}$, so equation 5 can be simplified into*

$$\mathcal{L}_{\mathcal{P}} = \mathbb{E}_{\mathbf{h} \sim \mathcal{P}_{\mathbf{H}}} \left[ V_{\mu_{\mathbf{h}}}^{\mathrm{TV}} + c \cdot V_{\mu_{\mathbf{h}}}^{\mathrm{TD}} \right], \tag{6}$$

*where $\mathcal{P}_{\mathbf{H}}$ is a selection distribution of $\mathbf{h} \in \mathbf{H} = \{\mathbf{h}_1, ..., \mathbf{h}_N\}$. For each actor, the behavior is generated from a selected policy model $\Phi_{\mathbf{h}_i}$ with a pre-defined behavior mapping $\mathcal{F}_{\psi}$.*

In Proposition 1, the behavior space size is controlled by the policy model population size (*i.e.*, $|\mathbf{H}|$). However, maintaining a large population of different policy models is data-consuming. Hence, we try to control behaviors via optimizing the selection of behavior mappings:

**Proposition 2** (Behavior Mapping Optimization). *When all the policy models are used to generate each behavior, e.g., $\mu_{\psi} = \mathcal{F}_{\psi}(\Phi_{\boldsymbol{\theta},\mathbf{h}})$ for single policy model cases or $\mu_{\psi} = \mathcal{F}_{\psi}(\Phi_{\boldsymbol{\theta}_1,\mathbf{h}_1}, ..., \Phi_{\boldsymbol{\theta}_N,\mathbf{h}_N})$ for N policy models cases, each behavior can be uniquely indexed by $\mathcal{F}_{\psi}$, and equation 5 can be simplified into:*

$$\mathcal{L}_{\mathcal{P}} = \mathbb{E}_{\psi \sim \mathcal{P}_{\Psi}} \left[ V_{\mu_{\psi}}^{\mathrm{TV}} + c \cdot V_{\mu_{\psi}}^{\mathrm{TD}} \right], \tag{7}$$

*where $\mathcal{P}_{\Psi}$ is a selection distribution of $\psi \in \Psi$.*

In Proposition 2, the behavior space is majorly controlled by $|\Psi|$, which could be a continuous parameter space. Hence, a larger behavior space can be enabled.

**Selection Distribution Optimization**    Given the optimization target $\mathcal{L}_{\mathcal{P}}$, we seek to find the optimal behavior selection distribution $\mathcal{P}_{\mu}^{*}$ that maximizes $\mathcal{L}_{\mathcal{P}}$:

$$
\mathcal{P}_{\mathbf{M}_{\mathbf{H},\Psi}}^{*} := \underset{\mathcal{P}_{\mathbf{M}_{\mathbf{H},\Psi}}}{\operatorname{argmax}} \mathcal{L}_{\mathcal{P}} \overset{(1)}{=} \underset{\mathcal{P}_{\mathbf{H}}}{\operatorname{argmax}} \mathcal{L}_{\mathcal{P}}
$$
$$
\overset{(2)}{=} \underset{\mathcal{P}_{\Psi}}{\operatorname{argmax}} \mathcal{L}_{\mathcal{P}}, \tag{8}
$$

where (1) and (2) hold because we have Proposition 1 and 2, respectively. This optimization problem can be solved with existing optimizers, *e.g.*, evolutionary algorithm (Jaderberg et al., 2017), multi-arm bandits (MAB) (Badia et al., 2020a), etc.

## 4    LBC-BM: A Boltzmann Mixture based Implementation for LBC

In this section, we provide an example of improving the behavior control of off-policy actor-critic methods (Espeholt et al., 2018) via optimizing the behavior mappings as Proposition 2. We provide a practical design of hybrid behavior mapping, inducing an implementation of LBC, which we call **B**oltzmann **M**ixture based **LBC**, namely LBC-$\mathcal{BM}$. By choosing different $\mathbf{H}$ and $\Psi$, we can obtain a family of implementations of LBC-$\mathcal{BM}$ with different behavior spaces (see Sec. 5.4).

### 4.1    Boltzmann Mixture Based Behavior Space Construction

In this section, we provide a general hybrid behavior mapping design including three sub-processes:

**Generalized Policy Selection**    In Agent57, behavior control is achieved by selecting a single policy from the policy population at each iteration. Following this idea, we generalize the method to the case where multiple policies can be selected. More specifically, we introduce a importance weights vector $\boldsymbol{\omega}$ to describe how much each policy will contribute to the generated behavior, $\boldsymbol{\omega} = [\omega_1, ..., \omega_{\mathrm{N}}], \omega_i \geq 0, \sum_{i=1}^{\mathrm{N}} \omega_i = 1$, where $\omega_i$ represents the importance of $i$th policy in the population (*i.e.*, $\Phi_{\mathbf{h}_i}$). In particular, if $\boldsymbol{\omega}$ is a one-hot vector, *i.e.*, $\exists i \in \{1, 2, ..., \mathrm{N}\}, \omega_i = 1; \forall j \in \{1, 2, ..., \mathrm{N}\} \neq i, \omega_j = 0$, then the policy selection becomes a single policy selection as Proposition 1. Therefore, it can be seen as a generalization of single policy selection, and we call this process *generalized policy selection*.

**Policy-Wise Entropy Control**    In our work, we propose to use entropy control (which is typically rule-based controlled in previous works) to make a better trade-off between exploration and exploitation. For a policy model $\Phi_{\mathbf{h}_i}$ from the population, we will apply a entropy control function $f_{\tau_i}(\cdot)$, *i.e.*, $\pi_{\mathbf{h}_i, \tau_i} = f_{\tau_i}(\Phi_{\mathbf{h}_i})$, where $\pi_{\mathbf{h}_i, \tau_i}$ is the new policy after entropy control, and $f_{\tau_i}(\cdot)$ is parameterized by $\tau_i$. Here we should note that the entropy of all the policies from the population is controlled in a policy-wise manner. Thus there would be a set of entropy control functions to be considered, which is parameterized by $\boldsymbol{\tau} = [\tau_1, ..., \tau_{\mathrm{N}}]$.

**Behavior Distillation from Multiple Policies**    Different from previous methods where only one policy is used to generate the behavior, in our approach, we combine N policies $[\pi_{\mathbf{h}_1, \tau_1}, ..., \pi_{\mathbf{h}_{\mathrm{N}}, \tau_{\mathrm{N}}}]$, together with their importance weights $\boldsymbol{\omega} = [\omega_1, ..., \omega_{\mathrm{N}}]$. Specially, in order to make full use of these policies according to their importance, we introduce a *behavior distillation function $g$* which takes both the policies and importance weights as input, *i.e.*, $\mu_{\mathbf{H}, \boldsymbol{\tau}, \boldsymbol{\omega}} = g(\pi_{\mathbf{h}_1, \tau_1}, ..., \pi_{\mathbf{h}_{\mathrm{N}}, \tau_{\mathrm{N}}}, \boldsymbol{\omega})$. The distillation function $g(\cdot, \boldsymbol{\omega})$ can be implemented in different ways, *e.g.*, knowledge distillation (supervised learning), parameters fusion, etc. In conclusion, the behavior space can be constructed as,

$$
\mathbf{M}_{\mathbf{H}, \Psi} = \{g\left(f_{\tau_1}(\Phi_{\mathbf{h}_1}), ..., f_{\tau_{\mathrm{N}}}(\Phi_{\mathbf{h}_{\mathrm{N}}}), \omega_1, ..., \omega_{\mathrm{N}}\right) | \psi \in \Psi\} \tag{9}
$$

wherein $\Psi = \{\psi = (\tau_1, ..., \tau_{\mathrm{N}}, \omega_1, ..., \omega_{\mathrm{N}})\}$, $\mathbf{H} = \{\mathbf{h}_1, ..., \mathbf{h}_{\mathrm{N}}\}$. Note that this is a general approach which can be applied to different tasks and algorithms by simply selecting different entropy control function $f_{\tau_i}(\cdot)$ and behavior distillation function $g(\cdot, \boldsymbol{\omega})$. As an example, for Atari task, we model the policy as a Boltzmann distribution, *i.e.*, $\pi_{\mathbf{h}_i, \tau_i}(a|s) = e^{\tau_i \Phi_{\mathbf{h}_i}(a|s)} \sum_{a'} e^{\tau_i \Phi_{\mathbf{h}_i}(a'|s)}$, where $\tau_i \in (0, \infty)$. The entropy can thus be controlled by controlling the temperature. As for the behavior distillation function, we are inspired by the behavior design of GDI, which takes a weighted sum of two softmax distributions derived from two advantage functions. We can further extend this approach

to the case to do a combination across different policies, *i.e.*, $\mu_{\mathbf{H},\boldsymbol{\tau},\boldsymbol{\omega}}(a|s) = \sum_{i=1}^{N} \omega_i \pi_{\mathbf{h}_i,\tau_i}(a|s)$. This formula is actually a form of *mixture model*, where the importance weights play the role of mixture weights of the mixture model. Then the behavior space becomes,

$$\mathbf{M}_{\mathbf{H},\boldsymbol{\Psi}} = \{\mu_{\mathbf{H},\boldsymbol{\psi}} = \sum_{i=1}^{N} \omega_i \operatorname{softmax}_{\tau_i}(\Phi_{\mathbf{h}_i}) | \boldsymbol{\psi} \in \boldsymbol{\Psi}\} \tag{10}$$

### 4.2 MAB BASED BEHAVIOR SELECTION

According to Proposition 2, the behavior selection over behavior space 10 can be simplified to the selection of $\boldsymbol{\psi}$. In this paper, we use MAB-based meta-controller to select $\boldsymbol{\psi} \in \boldsymbol{\Psi}$. Since $\boldsymbol{\Psi}$ is a continuous multidimensional space, we discretize $\boldsymbol{\Psi}$ into K regions $\{\Psi_1, ..., \Psi_K\}$, and each region corresponds to an arm of MAB. At the beginning of a trajectory $i$, $l$-th actor will use MAB to sample a region $\Psi_k$ indexed by arm k according to $\mathcal{P}_{\Psi} = \operatorname{softmax}(\operatorname{Score}_{\Psi_k}) = \frac{e^{\operatorname{Score}_{\Psi_k}}}{\sum_j e^{\operatorname{Score}_{\Psi_j}}}$. We adopt UCB score as $\operatorname{Score}_{\Psi_k} = V_{\Psi_k} + c \cdot \sqrt{\frac{\log(1 + \sum_{j \neq k}^{K} N_{\Psi_j})}{1 + N_{\Psi_k}}}$ to tackle the reward-diversity trade-off problem in equation 7 (Garivier & Moulines, 2011). $N_{\Psi_k}$ means the number of the visit of $\Psi_k$ indexed by arm $k$. $V_{\Psi_k}$ is calculated by the expectation of the undiscounted episodic returns to measure the value of each $\Psi_k$, and the UCB item is used to avoid selecting the same arm repeatedly and ensure sufficient diverse behavior mappings can be selected to boost the behavior diversity. After an $\Psi_k$ is sampled, a $\boldsymbol{\psi}$ will be uniformly sampled from $\Psi_k$, corresponding to a behavior mapping $\mathcal{F}_{\boldsymbol{\psi}}$. With $\mathcal{F}_{\boldsymbol{\psi}}$, we can obtain a behavior $\mu_{\boldsymbol{\psi}}$ according to equation 10. Then, the $l$-th actor acts $\mu_{\boldsymbol{\psi}}$ to obtain a trajectory $\tau_i$ and the undiscounted episodic return $G_i$, then $G_i$ is used to update the reward model $V_{\Psi_k}$ of region $\Psi_k$ indexed by arm $k$. As for the nonstationary problem, we are inspired from GDI, which ensembles several MAB with different learning rates and discretization accuracy. We can extend to handle the nonstationary problem by jointly training a population of bandits from very exploratory to purely exploitative (i.e., different c of the UCB item, similar to the policy population of Agent57). Moreover, we will periodically replace the members of the MAB population to ease the nonstationary problem further. More details of implementations of MAB can be found in App. E. Moreover, the mechanism of the UCB item for behavior control has not been widely studied in prior works, and we will demonstrate how it boosts behavior diversity in App. K.3.

## 5 EXPERIMENT

In this section, we design our experiment to answer the following questions:

- Whether our methods can outperform prior SOTA RL algorithms in both sample efficiency and final performance in Atari 1B Benchmarks (See Sec. 5.2 and Figs. 3)?
- Can our methods adaptively adjust the exploration-exploration trade-off (See Figs. 4)?
- How to enlarge or narrow down the behavior space? What is the performance of methods with different behavior spaces (See Sec. 5.4)?
- How much performance will be degraded without proper behavior selection (See Figs. 5)?

### 5.1 EXPERIMENTAL SETUP

#### 5.1.1 EXPERIMENTAL DETAILS

We conduct our experiments in ALE (Bellemare et al., 2013). The standard pre-processing settings of Atari are identical to those of Agent57 (Badia et al., 2020a), and related parameters have been concluded in App. I. We employ a separate evaluation process to record scores continuously. We record the undiscounted episodic returns averaged over five seeds using a windowed mean over 32 episodes. To avoid any issues that aggregated metrics may have, App. J provides full learning curves for all games and detailed comparison tables of raw and normalized scores. Apart from the mean and median HNS, we also report how many human worlds records our agents have broken to emphasize the superhuman performance of our methods. For more experimental details, see App. H.

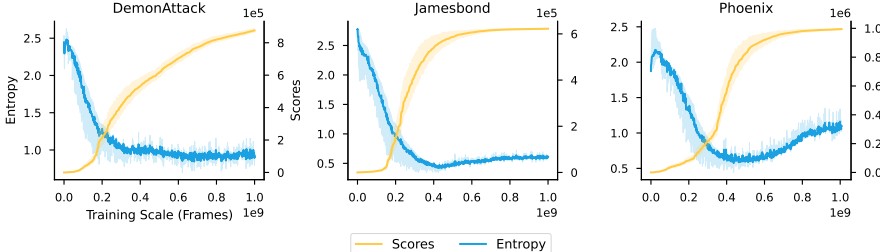

Figure 3: The learning curves in Atari. Curves are smoothed with a moving average over 5 points.

Figure 4: Behavior entropy and scores curve across training for different games where we achieved unprecedented performance. The names of the axes are the same as that of the leftmost figure.

### 5.1.2 IMPLEMENTATION DETAILS

We jointly train three polices, and each policy can be indexed by the hyper-parameters $\mathbf{h}_i = (\gamma_i, \mathcal{RS}_i)$, wherein $\mathcal{RS}_i$ is a reward shaping method (Badia et al., 2020a), and $\gamma_i$ is the discounted factor. Each policy model $\Phi_{\mathbf{h}_i}$ adopts the dueling network structure (Wang et al., 2016), where $\Phi_{\mathbf{h}_i} = A_{\mathbf{h}_i} = Q_{\mathbf{h}_i} - V_{\mathbf{h}_i}$. More details of the network structure can be found in App. L. To correct for harmful discrepancy of off-policy learning, we adopt V-Trace (Espeholt et al., 2018) and ReTrace (Munos et al., 2016) to learn $V_{\mathbf{h}_i}$ and $Q_{\mathbf{h}_i}$, respectively. The policy is learned by policy gradient (Schulman et al., 2017). Based on equation 10, we could build a behavior space with a hybrid mapping as $\mathbf{M}_{\mathbf{H},\mathbf{\Psi}} = \{\mu_{\mathbf{H},\psi} = \sum_{i=1}^{3} \omega_i \operatorname{softmax}_{\tau_i}(\Phi_{\mathbf{h}_i})\}$, wherein $\mathbf{H} = \{\mathbf{h}_1, \mathbf{h}_2, \mathbf{h}_3\}$, $\mathbf{\Psi} = \{\psi = (\tau_1, \omega_1, \tau_2, \omega_2, \tau_3, \omega_3) | \tau_i \in (0, \tau^+), \sum_{j=1}^{3} \omega_j = 1\}$. The behavior selection is achieved by MAB described in Sec. 4.2, and more details can see App. E. Finally, we could obtain an implementation of LBC-$\mathcal{BM}$, which is our **main algorithm**. The target policy for $A_1^\pi$ and $A_2^\pi$ in GDI-H$^3$ is the same, while in our work the target policy for $A_i^{\pi_i}$ is $\pi_i = \operatorname{softmax}(A_i)$.

### 5.2 SUMMARY OF RESULTS

**Results on Atari Benchmark** The aggregated results across games are reported in Figs. 3. Among the algorithms with superb final performance, our agents achieve the best mean HNS and surpass the most human world records across 57 games of the Atari benchmark with relatively minimal training frames, leading to the best learning efficiency. Noting that Agent57 reported the maximum scores across training as the final score, and if we report our performance in the same manner, our median is 1934%, which is **higher** than Agent57 and demonstrates our superior performance.

**Discussion of Results** With LBC, we can understand the mechanisms underlying the performance of GDI-H$^3$ more clearly: **i)** GDI-H$^3$ has a high-capacity behavior space and a meta-controller to optimize the behavior selection **ii)** only a single target policy is learned, which enables stable learning and fast converge (See the case study of KL divergence in App. N). Compared to GDI-H$^3$, to ensure the behavior space will not degenerate, LBC maintains a population of diverse policies and, as a price, sacrifices some sample efficiency. Nevertheless, LBC can **continuously** maintain a significantly larger behavior space with hybrid behavior mapping, which enables RL agents to continuously explore and get improvement.

### 5.3 CASE STUDY: BEHAVIOR CONTROL

To further explore the mechanisms underlying the success of behavior control of our method, we adopt a case study to showcase our control process of behaviors. As shown in Figs. 4, in most tasks,

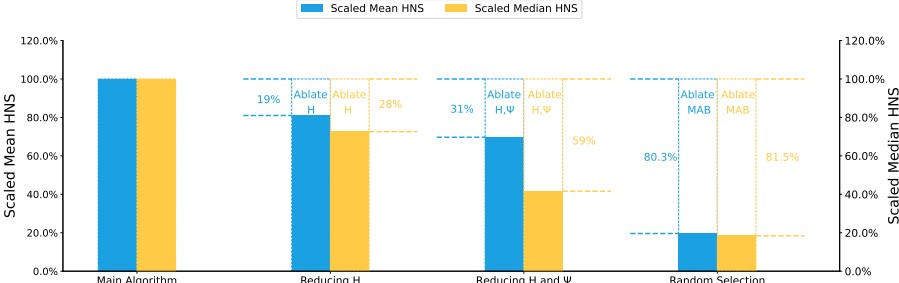

Figure 5: Ablation Results. All the results are scaled by the main algorithm to improve readability.

our agents prefer exploratory behaviors first (i.e., high stochasticity policies with high entropy), and, as training progresses, the agents shift into producing experience from more exploitative behaviors. On the verge of peaking, the entropy of the behaviors could be maintained at a certain level (task-wise) instead of collapsing swiftly to zero to avoid converging prematurely to sub-optimal policies.

## 5.4 ABLATION STUDY

In this section, we investigate several properties of our method. For more details, see App. K.

**Behavior Space Decomposition**   To explore the effect of different behavior spaces, we decompose the behavior space of our main algorithm via reducing $\mathbf{H}$ and $\mathbf{\Psi}$:

**1) Reducing H.** When we set all the policy models of our main algorithm the same, the behavior space transforms from $\mathcal{F}(\Phi_{\mathbf{h}_1}, \Phi_{\mathbf{h}_2}, \Phi_{\mathbf{h}_3})$ into $\mathcal{F}(\Phi_{\mathbf{h}_1}, \Phi_{\mathbf{h}_1}, \Phi_{\mathbf{h}_1})$. $\mathbf{H}$ degenerates from $\{\mathbf{h}_1, \mathbf{h}_2, \mathbf{h}_3\}$ into $\{\mathbf{h}_1\}$. We can obtain a control group with a smaller behavior space by reducing $\mathbf{H}$.

**2) Reducing H and $\mathbf{\Psi}$.** Based on the control group reducing $\mathbf{H}$, we can further reduce $\mathbf{\Psi}$ to further narrow down the behavior space. Specially, we can directly adopt a individual behavior mapping to build the behavior space as $\mathbf{M}_{\mathbf{H}, \mathbf{\Psi}} = \{\mu_\psi = \mathrm{softmax}_\tau(\Phi_{\mathbf{h}_1})\}$, where $\mathbf{\Psi}$ degenerates from $\{\boldsymbol{\omega}_1, \boldsymbol{\omega}_2, \boldsymbol{\omega}_3, \tau_1, \tau_2, \tau_3\}$ to $\{\tau\}$ and $\mathbf{H} = \{\mathbf{h}_1\}$. Then, we can obtain a control group with the smallest behavior space by reducing $\mathbf{H}$ and $\mathbf{\Psi}$.

The performance of these methods is illustrated in Figs. 5, and from left to right, the behavior space of the first three algorithms decreases in turn (According to Corollary 4 in App. C). It is evident that narrowing the behavior space via reducing $\mathbf{H}$ or $\mathbf{\Psi}$ will degrade the performance. On the contrary, the performance can be boosted by enlarging the behavior space, which could be a promising way to improve the performance of existing methods.

**Behavior Selection**   To highlight the importance of an appropriate behavior selection, we replace the meta-controller of our main algorithm with a random selection. The ablation results are illustrated in Figs. 5, from which it is evident that, with the same behavior space, not learning an appropriate selection distribution of behaviors will significantly degrade the performance. We conduct a t-SNE analysis in App. K.3 to demonstrate that our methods can acquire more diverse behaviors than the control group with pre-defined behavior mapping. Another ablation study that removed the UCB item has been conducted in App. K.3 to demonstrate the behavior diversity may be boosted by the UCB item, which can encourage the agents to select more different behavior mappings.

## 6 CONCLUSION

We present the first deep reinforcement learning agent to break 24 human world records in Atari using only 1B training frames. To achieve this, we propose a general framework called LBC, which enables a significantly enlarged behavior selection space via formulating a hybrid behavior mapping from all policies, and constructs a unified learnable process for behavior selection. We introduced LBC into off-policy actor-critic methods and obtained a family of implementations. A large number of experiments on Atari have been conducted to demonstrate the effectiveness of our methods empirically. Apart from the full results, we do detailed ablation studies to examine the effectiveness of the proposed components. While there are many improvements and extensions to be explored going forward, we believe that the ability of LBC to enhance the control process of behaviors results in a powerful platform to propel future research.

ACKNOWLEDGMENTS

This work is majorly supported by the National Key R&D Program of China (Grant Number 2021ZD0110400). In addition, this work is partly supported by the National Natural Science Foundation of China under Grant 62171248, and the R&D Program of Shenzhen under Grant JCYJ20220818101012025.

We are very grateful for the careful reading and insightful reviews of meta-reviewers and reviewers.

REPRODUCIBILITY STATEMENT

Open-sourced code will be implemented with Mindspore (MS, 2022) and released on our website.

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
