# OpenReview forum: "Learnable Behavior Control: Breaking Atari Human World Records via Sample-Efficient Behavior Selection"
_ICLR.cc/2023/Conference — ICLR 2023 notable top 5%_

### Official Review · Reviewer_PtKB · 2022-10-23

**Confidence:** 5
**Correctness:** 3
**Technical Novelty And Significance:** 2
**Empirical Novelty And Significance:** 2
**Recommendation:** 10

**Clarity, Quality, Novelty And Reproducibility:**

The clarity of the submission is good.

The quality of the submission is very high.

The novelty of the submission is low. (See comments above.)

The reproducibility of the submission is low. (I think that attempting to exactly reproduce the results without any code would require an unreasonable effort in this case.) **The value of the submission would be substantially increased if code could be released.**

**Strength And Weaknesses:**

# Main Concern

This situation is unfortunate. **The submission is excellent** (simple approach, great empirical results, nice writing), and the authors clearly spent a large amount of time to make it as good as it is. **But there is existing work that was arxived (https://arxiv.org/abs/2106.06232v1) more than a year ago and recently published in ICML (https://proceedings.mlr.press/v162/fan22c) that proposed essentially the same methodology and achieves similar empirical results.** I will refer to this existing work as GDI.

### Irresponsibility of the omission

GDI was published by a relatively lesser known research group among the reinforcement learning community and has not received as much attention as it deserves, so I can see how the authors may not have known about it. However, even if this omission was unintentional, I feel it was still irresponsible: When making a submission with claims of novelty, the submitters are staking their credibility on the claims of novelty being factual; the submitters appear to have made such claims without having done due diligence.

### Work is not concurrent

I do not think the submission can claim concurrency with GDI since GDI has been on arxiv since June 2021.

### Similarity of LBC and GDI implementations

In their implemented algorithms, both works use continuous MAB algorithms to optimize the temperatures of softmax functions and mixture parameters over policies. There are many instantiation-level differences, but I feel that the implementation in the submission can meaningfully be described as an instance of GDI (and vice versa). If the authors disagree with this characterization, I ask that they provide detailed evidence for their claims.

### Similarity of empirical results

Across various metrics, the submission reports results (e.g., 10077.52% mean human normalized score and 24 human world records) that are a bit better than GDI (e.g., 9620.98% mean human normalized score and 22 human world records) after 5 times more samples (1B for LBC vs. 200M for GDI). It is hard to make a precise comparison, but, if anything, I might say that GDI's empirical results are a bit more impressive, given that it achieved almost as good results with 5x fewer samples.

### Value of submission

Although I feel that there is not algorithmic novelty and that the empirical results are not substantially better, I think the submission as is offers value in the sense that it is much more clearly written than GDI. Since the idea behind GDI/LBC is, in my view, quite important for modern RL but not currently widely understood by the community, I think it deserves to have a paper where it is explained more clearly than it was in GDI. That said, I do not think it would be fair to the authors of GDI to re-brand the same idea under a different name and merely mention GDI as related work.

## Other thoughts

### Measurement notation

The submission chooses to break the optimization target into two terms that it calls a diversity-based measurement and a value-based measurement. I am not sure what purpose doing this serves, as the submission never actually comes back to this point when discussing the actual algorithm it implemented. I think the best way to resolve the issue would be to use a single symbol V to refer to a measurement and note in the text that this measurement may be decomposable into some weighted combination of diversity and value. Alternatively, if the authors do not want to do that, I think they should at least describe the algorithm that they implemented in terms of these two measurements at some point in the submission so that readers are able to make the connection.

### "Goal-directed" terminology

The submission repeatedly uses the terminology "goal-directed". To me, this feels like unnecessary jargon. The submission suggests that Agent57 uses a goal-directed meta controller, but I could not find any usage of "goal-directed" in the Agent57 paper. If the submission does not want to remove this terminology, I think the submission should at least make it clear exactly what it means by "goal-directed" and how this usage departs from or is the same as that in existing reinforcement learning literature.

### EfficientZero final performance

The submission states:
> From Figs. 3, EfficientZero (Ye et al., 2021) achieves remarkable learning efficiency in smaller data volumes but fails to obtain comparable final performance as other SOTA methods like Agent57 or Muzero.

This statement may be superficially true, but it is misleading to readers because it is comparing apples to oranges. The numbers reported in the submission for LBC use **four orders of magnitude!** more samples than EfficientZero. It is entirely plausible that the final performance of EfficientZero would be comparable to that of MuZero given the same number of samples. Thus, saying that it does not achieve the same "final performance" does not strike me as appropriate.

On this note, I do not think it is even necessary to include EfficientZero(100k) in these plots. It is not a relevant baseline for the submission due to the drastically different sample usages.

## Closing Thoughts

As articulated above, I feel that the submission is a great piece of work and, even given existing work, offers value to the community. I hope that, through conversing with the authors, we will be able to agree on appropriate revisions that will make the submission acceptance-worthy. However, I anticipate that these changes may be quite substantial -- i.e., going well beyond simply mentioning GDI as related work. I think they would likely require the submission to be re-written from the perspective of providing additional clarity and empirical confirmation for GDI, rather than that of proposing a novel methodology. To me, it is important that the authors of GDI not be denied credit for having had an important idea simply because they are not well-known RL researchers.

**Summary Of The Paper:**

The submission proposes a mechanism that it calls learnable behavioral control (LBC). LBC aggregates the policies of a population into a single policy using a bandit-based meta controller. The submission shows that LBC achieves very strong results on Atari.

**Summary Of The Review:**

The main methodological contribution of the submission has already been proposed by existing work, which was first arxived over a year ago and recently published in ICML; furthermore, this existing work achieves similar empirical results to the submission. Because the submission does not acknowledge this existing work, and instead presents its methodology as a novel contribution, I do not think it is acceptable in its current form.

---

After discussion with the authors and revisions of the submission, I raised my score.

---

### Official Review · Reviewer_Spxv · 2022-10-24

**Confidence:** 4
**Correctness:** 4
**Technical Novelty And Significance:** 3
**Empirical Novelty And Significance:** 3
**Recommendation:** 8

**Clarity, Quality, Novelty And Reproducibility:**

Quality

- Paper proposes an interesting and novel idea for improving sample efficiency by increasing the behavior diversity without increasing the population size.
- The experimental results show the effectiveness.
- It’s unclear what categories of Atari games get the most improvement and which games didn’t not get improvements, can the authors add a discussion on this? You could categorize the games according to their difficulty.

Clarity
- Mostly very clear. Some implementation details are missing.
- The goal-directed term is used many times throughout the paper without explanation or definition except a reference to Agent57’s paper,  please address this in the next version.
- It would be great to add a discussion on the difference between this work and MEME.
Please consider referring to the per game score that is only available in the supplementary material.


**Strength And Weaknesses:**

- Paper seems well written, key concepts are explained, and claims are supported.

- The results are very good. As many games of the ALE benchmark require exploration, the significantly improved results show that the method is highly effective at exploring. I'm impressed by the effectiveness of this approach.

- The behavior construction and selection method is novel to me although seems too complicated to get it to work well in practice. But the authors provide important hyperparameters and pseudo algorithms. I think this is a good work that demonstrates advances of exploration leads to unprecedented performance on Atari games.


**Summary Of The Paper:**

This work considers the exploration problem in RL. Building upon prior work on population based training, this work aims to increase behavior space without increasing the population size in order to improve sample efficiency.
The method is hybrid behavior mapping which learns mapping functions to get sampling behaviors from policies. Experimental results show this method achieves new state-of-the-art performance on the ALE benchmark.


**Summary Of The Review:**

This paper proposes a novel and effective exploration method that achieves good results on Atari games.
There are some actionable non-critical issues that I hope could be addressed in the next version.

---

### Official Review · Reviewer_qVG5 · 2022-10-26

**Confidence:** 4
**Correctness:** 4
**Technical Novelty And Significance:** 3
**Empirical Novelty And Significance:** 4
**Recommendation:** 8

**Clarity, Quality, Novelty And Reproducibility:**

**Clarity:**
- As mentioned above, clarity is the biggest weakness of the paper. I strongly encourage the authors to clarify Section 4.2 to more precisely explain how the MAB approach selects multiple polices to create the mixture model. If the authors could comment on this in the rebuttal that would also be helpful.
- It is a bit unconventional to put a key results plot in the abstract itself.
- Explain the acronym NGU
- Section 2.2 is oddly placed, since it repeatedly mentions behavior control and behavior mapping before those terms are explained. It does not contribute much to the understanding of the paper at this point.
- Clarify whether the ablation results in Figure 5 apply all of the ablations to each successive item on the x-axis (as in, for "Random Selection", are H and $\Psi$ also ablated for this experiment?)

**Quality:**
- As mentioned above, the results are potentially highly significant and of interest to the community.
- Assumption 1 is generally untrue... different random initializations of the network weights, even for the same hyperparameters, can lead to significantly different policies. Why is it necessary to formalize Assumption 1 rather than simply introduce Equation 4 once and use a footnote to mention you leave out $\theta$ as a notational convenience?

**Novelty:**
- Many components of the proposed approach have been introduced in prior work, i.e. using PBRL in Agent57 and using a MAB to select from the population in Badia et al. (2020).
- The mixture policy created in this work and the proposed "Generalized Policy Selection" is reminiscent of Generalized Policy Improvement in Successor Features https://arxiv.org/abs/1606.05312 https://www.pnas.org/doi/10.1073/pnas.1907370117. It would be interesting to explore that connection.

**Strength And Weaknesses:**

A key strength of the paper is in the significance of the results. Setting a new state-of-the-art in Atari is a significant accomplishment, and will be of interest to the RL community.

A weakness of the paper is in a lack of clarity around key technical details, combined with a lot of redundant and unnecessary explanation characterizing the space of possible techniques. The paper spends 3-4 pages (p. 3-6) on redundant and repetitive explanations of behavior space vs. behavior mapping. Many parts of these pages are repeated (for example, the "Behavior Formulation" paragraph on p.4 is fully subsumed in other parts of the section and does not need to be repeated). However, the paper spends relatively little time justifying or giving intuition for why the central contribution of the paper (Eq. 9) is the right approach. Most importantly, the explanation of how the MAB behavior selection interacts with Eq. 9 is left very unclear. It would seem that to apply Eq. 9 the MAB would need to select N policies, but this is not stated in the text, which instead says that Eq. 9 will be applied to a single $\Phi_k$, which does not make sense. This lack of precision makes the paper hard to replicate.

**Summary Of The Paper:**

The paper proposes a new technique for enhancing population-based RL (PBRL), and shows state-of-the-art results on the Atari Learning Environment (ALE). The technique expands upon previous work in PBRL, which used a multi-armed bandit (MAB) meta-controller to select the best policy from a population of policies. This paper also uses the MAB approach, but proposes to combine multiple policies from the population by summing their softmax outputs, creating a mixture model. This allows them to achieve the highest number of world records in Atari of any previous algorithm.

**Summary Of The Review:**

In summary, the paper should be accepted given that the results are impressive and are of significant interest to the community. However, the organization and technical communication in the paper could be significantly improved to enhance its potential impact and reproducibility.

If it were not for the issues with technical communication pointed out above, I would increase my score to indicate that the paper should be highlighted at the conference.

---

### Comment · Reviewer_PtKB · 2022-11-18
**Centralizing Conversation**

I am making a new reply here to centralize remaining discussions with the authors. I am making this response hastily in hopes that the authors have time for one more revision of the text before the end of the discussion period.

> A1: We are sorry for the unclear quotation. In sec 5.1, 2nd paragraph of GDI's ICML paper (https://proceedings.mlr.press/v162/fan22c), "The learning policy is also ". GDI-I3 and GDI-H3 hold exactly the same RL training process (except for different reward scales for  and ), and we can see that both and are learned from the same target policy. And for the "off-policy update", more precisely, importance sampling, is only used to handle the time delay of the asynchronous parallel actors.

I read through the section of the GDI paper referenced above. The "The learning policy is also " sentence seems to refer to GDI-I3; I can see how implicitly it may also refer to GDI-H3.

Though I am not sure it is entirely clear from the GDI text, I am satisfied with the authors' clarifications on this matter. I think part of the difficulty in making a determination here is the lack of writing clarity of the GDI paper, which clearly should not be held against the authors of the submission.

> A2: To demonstrate this, we supplemented a new set of case studies in App. K to demonstrate that with the same target policy, the KL divergence of the induced softmax distributions of  and  learned under the same reward shapings as GDI-H will gradually decrease close to zeros.

I apologize -- I think I was unclear before. I did not mean to ask the authors to perform additional experiments, though I certainly appreciate them doing so. Can the authors specify what kind of reward shaping was used? Is it the same kind that was used in GDI?

---

### Remaining changes for requested:

I think the current draft is much closer to an appropriate acknowledgement level than the previous draft. I thank the authors for their sincere engagement with the review process. Below I have listed remaining thoughts on how I think the submission could be improved with respect to its treatment of GDI:
1. In the introduction, the submission mentions GDI but immediately points to perceived drawbacks. **A more generous writing style would instead explain how the submission *builds upon* GDI, rather than detracting it**. *I think this is especially preferable since there is not a clear cut empirical takeaway on whether GDI or LBC achieves superior empirical performance.*
2. I think the compare and contrast with GDI in the behavior distillation section is great. **I ask that the authors add an analogous compare/contrast paragraph for the MAB BASED BEHAVIOR SELECTION section**.
3. Regarding the diagrammatic comparison with GDI in the appendix. **I ask that the authors change the caption to GDI-H3 to indicate the particular instantiation of GDI to which the diagram refers.**
4. Regarding Figure 12 in the appendix, **I ask that the authors specify the kind of reward shaping was used so that readers may better understand the experiment.**

I again thank the authors for their high level of engagement with the review process and hope that they will be able to make these final requested changes before the window for modifications closes.

---

### Comment · Reviewer_PtKB · 2023-09-27
**Official Implementation**

The authors committed to releasing their code both in the text (see Reproducibility statement) and in their rebuttal (see Response to Reviewer PtKB [4/4]). This commitment was an important factor in appraisal of the submission and it is important that the authors take their commitment seriously. Insofar as I can tell, there is no code available at this time. Can the authors please provide an update regarding the status of their code? The ICLR conference occurred nearly 5 months ago, decision came out over 8 months ago, and the authors made the commitment in their rebuttal over 10 months ago. It is well past the time that the code should've been released.

---

### Decision · Program_Chairs · 2023-01-20

**Decision:**

Accept: notable-top-5%

**Justification For Why Not Higher Score:**

N/A

**Justification For Why Not Lower Score:**

Very impressive results which will be of interest to the RL community. Reviewers further thought this paper would give exposure to the closely related Generalized Data Distribution Iteration (GDI) method which has not yet received as much attention from the community. An oral would be a great way to achieve this.

One reason for bumping down is the relatively low novelty with respect to GDI. The paper itself could also be significantly improved, with respect to clarity ("redundant and repetitive explanations of behavior space vs. behavior mapping", qVG5) and technical details on the multi-armed bandit.

To clarify the confidence level, I am interpreting "This decision can be bumped down" in the sense of being bumped down from "Oral" to "Spotlight".

**Metareview: Summary, Strengths And Weaknesses:**

The paper introduces Learnable Behavior Control (LBC), a novel population-based training method which achieves state-of-the-art results on the Atari benchmark, both in terms of human normalized score but (impressively) in human world records. Compared to prior work in this area, the method proposes to derive behavior policies from a *set* of distinct policies which are dynamically combined on the actors via a learnt behavioral mapping. Concretely, the authors propose to fuse these base policies via a mixture of tempered Boltzmann distributions. As in prior work, the parameters of the behavior mapping are adjusted online via a multi-armed bandit algorithm.

The reviewers overwhelmingly support acceptance of this paper. Results represent a “significant accomplishment” [qVG5], which “demonstrates [that] advances of exploration leads to unprecedented performance on Atari games” [SpxV]. All felt this paper would be of significant interest to the community.

The main point of contention during the discussion was the close relationship between LBC and Generalized Data Distribution Iteration (GDI). I am happy to see that this issue has now been resolved. [PtKB] and authors should both be commended for the constructive discussion. Overall, this has significantly improved the paper, both in terms of the framing of LBC (Section 2.2) but also in the comparison of empirical results (Fig. 3, Section 5.2, Fig 11, Appendix N). Other smaller issues of clarity and lack of technical details [qVG5] seemed to have been addressed during the rebuttal, though I would encourage the authors to further clarify the relationship between equation 6 and the UCB score introduced in Section 4.2, which personally only became clear after reading the reviewer discussion!

Given the complexity of the method, I sincerely hope the authors will follow through with their promise to open-source their implementation, alongside a dataset of generated trajectories. Congratulations to the authors on a great paper!

**Note From Pc:**

if the above contains the word "oral" or "spotlight" please see: "oral" presentation means -> notable-top-5% and "spotlight" means -> notable-top-25%. As stated in our emails, we are disassociating presentation type from AC recommendations

**Summary Of Ac-Reviewer Meeting:**

N/A